# Endotenon-Derived Type II Tendon Stem Cells Have Enhanced Proliferative and Tenogenic Potential

**DOI:** 10.3390/ijms242015107

**Published:** 2023-10-12

**Authors:** Marta Clerici, Vera Citro, Amy L. Byrne, Tina P. Dale, Aldo R. Boccaccini, Giovanna Della Porta, Nicola Maffulli, Nicholas R. Forsyth

**Affiliations:** 1School of Pharmacy and Bioengineering, Keele University, Stoke-on-Trent ST4 7QB, UK; m.clerici@keele.ac.uk (M.C.); v.citro@keele.ac.uk (V.C.); a.byrne1@keele.ac.uk (A.L.B.); t.p.dale@keele.ac.uk (T.P.D.); n.maffulli@qmul.ac.uk (N.M.); 2Department of Medicine, Surgery and Dentistry, University of Salerno, Via S. Allende, 84081 Baronissi, Italy; gdellaporta@unisa.it; 3Institute for Biomaterials, Department of Materials Science and Engineering, Friedrich-Alexander-University of Erlangen-Nürnberg, 91058 Erlangen, Germany; aldo.boccaccini@fau.de; 4Interdepartmental Centre BIONAM, University of Salerno, Via Giovanni Paolo I, 84084 Fisciano, Italy; 5Department of Trauma and Orthopaedic Surgery, University Hospital “San Giovanni di Dio e Ruggi D’Aragona”, 84131 Salerno, Italy; 6Department of Trauma and Orthopaedics, Faculty of Medicine and Psychology, Sant’Andrea Hospital, Sapienza University, 00189 Rome, Italy; 7Vice Principals’ Office, University of Aberdeen, Kings College, Aberdeen AB24 3FX, UK

**Keywords:** TSCs, TNCs, tendon regeneration, tendinopathies, tenogenic markers

## Abstract

Tendon injuries caused by overuse or age-related deterioration are frequent. Incomplete knowledge of somatic tendon cell biology and their progenitors has hindered interventions for the effective repair of injured tendons. Here, we sought to compare and contrast distinct tendon-derived cell populations: type I and II tendon stem cells (TSCs) and tenocytes (TNCs). Porcine type I and II TSCs were isolated via the enzymatic digestion of distinct membranes (paratenon and endotenon, respectively), while tenocytes were isolated through an explant method. Resultant cell populations were characterized by morphology, differentiation, molecular, flow cytometry, and immunofluorescence analysis. Cells were isolated, cultured, and evaluated in two alternate oxygen concentrations (physiological (2%) and air (21%)) to determine the role of oxygen in cell biology determination within this relatively avascular tissue. The different cell populations demonstrated distinct proliferative potential, morphology, and transcript levels (both for tenogenic and stem cell markers). In contrast, all tendon-derived cell populations displayed multipotent differentiation potential and immunophenotypes (positive for CD90 and CD44). Type II TSCs emerged as the most promising tendon-derived cell population for expansion, given their enhanced proliferative potential, multipotency, and maintenance of a tenogenic profile at early and late passage. Moreover, in all cases, physoxia promoted the enhanced proliferation and maintenance of a tenogenic profile. These observations help shed light on the biological mechanisms of tendon cells, with the potential to aid in the development of novel therapeutic approaches for tendon disorders.

## 1. Introduction

Tendon is the unique connective tissue component of the musculoskeletal system that links muscle to bone, translating muscular contractions into joint motion or stabilization [1]. Tendon injuries and pathologies are increasing in relation to human lifestyles and are common in daily life and sports [2]. The hypovascular and hypocellular nature of tendon tissues limits their healing capacity and contributes to the loss of functionality and propensity to reinjury [3,4]. Tendon healing is a complex coordinated event governed by the active interactions between resident cells and chemical and mechanical external stimuli [5]. As effective treatments for tendon injuries are limited, the development of new treatments with higher reliability and efficacy is crucial for improving outcomes for patients and athletes [6].

Despite the considerable progress made in identifying several important players in tendon biology, neither the ontogeny of the tenogenic lineage nor an accurate analysis of the heterogeneous cell population resident in tendon tissue has been provided [7]. For this reason, it becomes of primary importance to unequivocally identify the role of resident or stem cells in the actuation of bioactive molecules such as growth factors or in the establishment of tenogenic markers [8,9]. Traditionally, tendons were believed to contain only tenocytes, which were responsible for the maintenance and repair of the tissue [10,11]. Resident cells, which possess multipotency, self-renewal, and clonogenicity properties, were identified as stem cells in animals [12] and humans [12,13]. Tendon stem cells (TSCs), commonly referred to as tendon stem/progenitor cells (TSPCs) considering their heterogeneity, exhibit varying differentiation potentials [14]. TSPCs are understood to be situated in a distinct niche consisting of a unique ECM [1]. This niche is unlike other known stem cell niches, such as the bulge niche for skin stem cells [15], the osteoblast niche for hematopoietic stem cells [16], and the perivascular niche for neural stem cells and bone marrow stromal stem cells [17,18], which are all mainly cell-based and regulated. The TSPC niche is instead described—given the great abundance of tendons in ECM components and the lower number of cells compared with other tissues—as mostly made and regulated by ECM components, such as collagens; large proteoglycans; and small, leucine-rich proteoglycans, which function as lubricators and organizers for collagen fibril assemblies [19,20]. Moreover, the perivascular niche may be a source of yet another type of local stem/progenitor cells, with cells located in the perivascular space of tendon tissues simultaneously expressing tendon and pericyte-associated marker genes [21]. Furthermore, various stem/progenitor cells are distributed regionally throughout the tendon, specifically in the paratenon (type I TSCs) and the endotenon (type II TSCs) [21,22]. The paratenon, the loose areolar connective tissue that surrounds tendons, is primarily composed of elastic fibrils, type I and type III collagen fibrils, and synovial cells that line its inner surface [19]. Since the paratenon is elastic, the tendon can move freely in relation to the tissues around it [7]. The endotenon is a thin reticular network of connective tissue inside the tendon that contains a well-developed crisscross pattern of collagen fibrils, which bind and invest the tendon fibers [23]. The endotenon allows fiber groups to glide over each other and transports lymphatics, blood arteries, and nerves to deeper regions of the tendon [11,19]. Between the paratenon and the endotenon, there is another thin layer of connective tissue, the epitenon, which binds fascicles together to make the tendon [11]. Even though tendon subpopulations of cells are not clearly identified by specific markers, their role in the early healing response to acute injuries and ectopic tissue formation is distinct from the role of terminally differentiated tenocytes [24].

Stem cells have been studied for their self-renewal and tissue regeneration capacities in vitro, and tendon stem cells are now considered an ideal cell source for tendon repair [25,26]. However, little research has been conducted on porcine tendon stem cells [6,27], despite the growing body of literature on their suitability to act as an alternative for organ transplantation given their similarities in anatomical (e.g., cardiovascular system [28]), physiological (e.g., renal function [29]), and metabolic (e.g., digestive physiology [30]) characteristics. Moreover, pigs have organs and tissues of comparable size and function to human organs in both infancy and adulthood [31]. However, all these advantages of the pig are dependent upon the development of a successful method of overcoming the major immune response mounted against any grafted pig organ [28].

Previous studies have compared distinct cell populations from rabbits [32], horses [33], mice [34], and pigs [27]; however, no reports, to the best of our knowledge, have isolated and characterized three distinct cell populations in two different oxygen concentrations (21% and 2%). Here, we sought to compare TSCs (type I and II) and TNCs isolated from conspecific (sus scrofa domesticus) Achilles tendons [6,35]. We analyzed and compared the isolation, differentiation potential, and biological characteristics of TSCs (type I and II) and TNCs, applying a range of techniques, including flow cytometry, RT-qPCR, and immunofluorescence. We further determined the effect of oxygen concentration on the dynamic transition between the cell lines, cultured in physiological (2%) and air (21%) oxygen concentrations. We determined that all tendon-derived cell populations shared features; reduction in aspect ratios with increased passage, multipotency, and immunophenotypes (positive expression of CD44 and CD90) and had some distinct characteristics; proliferative potential, morphology, and transcription levels of tenogenic and stem cell markers. Among all tendon-derived cell populations evaluated, TSC II exhibited the highest proliferative potential, with the shortest doubling time across different passages, the highest number of CD44- and CD90-positive cells, and the retention of a multipotent potential across increased passage, and they were superior in the maintenance of both tenogenic and stem marker transcript levels. Furthermore, physoxia promoted enhanced proliferation in all cell types explored and maintained transcript levels in all cell populations and TNMD protein expression in type II TSCs.

## 2. Results

### 2.1. Cell Isolation and Characterization

Achilles tendons were dissected (Figure 1a); the paratenon was removed, minced, and digested in type II collagenase overnight for TSC type I extraction; and the endotenon was minced followed by either overnight digestion for TSC type II extraction or alternatively explanted, with digestion, for TNC isolation. Following enzymatic digestion, 1.6 million type I TSCs and 1.5 million type II TSCs per pig trotter (average tendon size 14 × 4 × 2 cm^3^) were obtained for subsequent seeding (Figure 1b).

With the explant method, the minced tendon tissue was cultured in 2% and 21% O_2_, and TNC migration from tissue pieces was apparent after 1 week. No immediate morphological differences between TNCs cultured in different oxygen conditions were apparent, but physoxia promoted faster cell proliferation, which reached confluence faster (<3 weeks) than TNCs in 21% O_2_ (>3 weeks).

### 2.2. Tendon-Derived Cell Populations Displayed Distinct Proliferative Rates

Cell growth rate was measured for all cells (TNCs, TSCs I and II), cultured for 120 days in physiological and atmospheric oxygen conditions (2% and 21% O_2_). Type I TSCs in 21% O_2_ plateaued after 40 days of culture at 15 cumulative population doublings (CPDs) and displayed relatively limited growth in the days thereafter. Type I TSCs in 2% O_2_ displayed continuous proliferation for 70 days before entering a reduced growth plateau at 35 CPDs, with a 57% increase in proliferative potential compared with the plateau point in 21% O_2_. Type II TSCs in 21% O_2_ proliferated across 60 days before gradually entering the plateau phase at 30 CPDs. Type II TSCs in 2% O_2_ also entered a reduced growth plateau after 70 days of culture, ultimately achieving 40 CPDs in contrast to the 30 CPDs seen in 21% O_2_, a 25% increase in ultimate proliferative potential. TNCs in 21% O_2_ plateaued after 90 days of culture, displaying relatively limited growth, while the proliferation rate of TNCs in 2% O_2_ remained constant during the time of culture, achieving 19 CPDs in contrast to 11 CPDs in air, a 42% increase in proliferative potential. Taken together (Figure 2a), TSC I proliferation was substantially reduced in 21% O_2_ when compared with that achieved in 2% O_2_, whereas type II TSCs displayed only a marginal reduction in proliferation in 21% O_2_. TNCs displayed a much earlier proliferative arrest but, irrespective, also displayed an air-induced inhibition of their full proliferative potential. Therefore, we can conclude that type II TSCs, derived from the endotenon membrane, have less oxygen sensitivity than the other cell populations. Moreover, type II TSCs displayed the highest proliferative potential in both oxygen conditions, while TNCs were proliferatively restricted compared with the others. Enhanced proliferation in 2% O_2_ was observed in all instances.

Consistent with the above, type I TSCs in 21% O_2_ displayed a consistent population doubling time (PDT) until they plateaued, and the PDT became four-fold higher, while in 2% O_2_, this was consistent during the time in culture (Figure 2b). Type II TSC PDTs in both oxygen concentrations were steady across the 120 days of culture. TNC PDTs in 21% O_2_ were consistent with TNCs in 2% O_2_ at early passages before entering the plateau and becoming four-fold higher than 2% O_2_. We can determine that type II TSCs displayed the shortest doubling time and that TNCs displayed the longest doubling time, regardless of the O_2_ concentration, while type I TSCs in 2% O_2_ showed a PDT close to type II TSCs and in 21% O_2_ a PDT closer to the behavior of TNCs.

### 2.3. Tendon-Derived Cell Population Aspect Ratio Reduced with Proliferative Age

Primary cells isolated from tendon tissue adhered to tissue culture plastic within hours. We noted a morphological similarity between type I TSCs and TNCs with an elongated morphology in comparison with type II TSCs, which were smaller and had a shorter population doubling time (PDT) (Figure 2b).

To quantify these distinct morphologies, we adopted the aspect ratio measure. The aspect ratio provides a convenient approach to quantifying cell morphological characteristics: the overall trend for TNCs is to move from being long and thin to larger, flatter, and more rounded, as described in rabbits [36] and humans [37]. The TSC I aspect ratio in 21% O_2_ decreased between passages 1 and 13, accompanied by a morphology shift from spindle-shaped to rounded (*p* = 0.0007), while no change was observed in type I TSCs in 2% O_2_ with increased passage number (Figure 3). The TSC II aspect ratio reduced between the first and last passage irrespective of condition, P19 in 21% O_2_ (*p* = 0.0077) and P20 in 2% O_2_ (*p* < 0.0001). Significant differences were noted between the aspect ratios of type II TSCs, both at early and late passages, cultured in differing oxygen conditions (Figure 4). TNC aspect ratios decreased over time between P1 and P10 in 21% O_2_ (*p* = 0.0008), P1 and P11 in 2% O_2_ (*p* < 0.0001), and the oxygen concentration used to culture the cells (Figure 5).

Therefore, the TSC I aspect ratio was sensitive to air oxygen (21%) with increasing passage. On the other hand, type II TSCs and TNCs not only showed a significant change in the aspect ratio following progressive passage but also in their oxygen sensitivity.

### 2.4. Tendon-Derived Cell Populations Displayed Multipotency Potential

To determine the differentiation potential of our three tendon-derived cell populations, we performed trilineage differentiation by using adipogenesis, osteogenesis, and chondrogenesis-inducing media. Type I TSCs displayed multipotency potential in early passage cells, which stained positively for histological stains (Oil red O (fat), Alizarin red (bone), and Alcian blue (cartilage)) following 21 days of exposure to differentiation media (Figure 6). For type II TSCs, multipotency was tested in both early (P3) (Figure 7a) and late passage (Figure 7b) populations. In both scenarios, positive histological staining was identified, confirming the presence and retention of a trilineage differentiation potential. Type II TSCs were selected for late-passage trilineage differentiation determination as the only tendon-derived cell population that displayed consistent long-term proliferative potential. Finally, TNCs also displayed multipotency potential in early passage cells treated with histologically identified trilineage differentiation (Figure 8). Visual inspection suggested that low efficiency of differentiation into adipocytes and chondrocytes in both oxygen concentrations was obtained compared with type I TSCs and early-passage type II TSCs. Consequently, we can determine that all tendon-derived cell populations have multipotent differentiation potential and that type II TSCs retain that potential in late-passage populations. Physoxia-differentiated populations, mostly adipocytes and chondrocytes, began detaching from the wells after 21 days of culture in 2% O_2_.

### 2.5. Tendon-Derived Cell Populations Displayed Similar Immunophenotypes

Following on from our demonstration that tendon-derived cell types all displayed a minimally tripotent differentiation potential, we next sought to determine whether they featured an immunophenotype profile consistent with ISCT guidelines for mesenchymal stromal/stem cells (MSCs) (e.g., negative for CD45, CD34, CD14, CD11b, CD79α, CD19, or HLA-DR and positive for CD73, CD90, and CD105) [38]. Therefore, we performed flow cytometry analysis to establish the expression levels of the following surface markers: CD14, CD19, CD34, CD44, CD45, HLA-DR, CD73, CD90, and CD105.

We first explored the expression of the hematopoietic markers, negative for MSCs, in all three of our tendon-derived cell populations. Type I TSCs displayed less than 0.5% positive cells for CD14, CD19, CD34, CD45, and HLA-DR. The same results were obtained for type II TSCs and TNCs (Figure 9).

We next explored the expression of positive markers for MSCs, CD44, CD90, CD73, and CD105, in all three of our tendon-derived cell populations. Type I TSCs displayed 60% CD44-positive cells in 21% O_2_ and over 75% in 2% O_2_. Type II TSCs in both oxygen conditions showed over 90% CD44-positive cells. Similar to type I TSCs, TNCs displayed a higher number of CD44-positive cells in 2% O_2_ (~80%) in contrast to 65% in 21% O_2_ (Figure 10a). Type I TSCs displayed a consistent fluorescence intensity in both oxygen conditions. Type II TSCs showed a 20% increase in atmospheric oxygen compared with physiological conditions. The TNCs’ mean fluorescence intensity in 2% O_2_ displayed a 17% increase over 21% O_2_ (Figure 10b). From this, we determined that, in both oxygen conditions, type II TSCs exhibited the highest number of CD44-positive cells, while TNCs displayed the highest fluorescence intensity.

Type I TSCs displayed 73% CD90-positive cells in 21% O_2_ in contrast to 87% in physiological oxygen concentrations. Like CD44, type II TSCs showed a consistent number of CD90-positive cells with less than a 5% difference between cells in 21 and 2% O_2_ (88% and 92%, respectively). TNCs displayed 94% CD90-positive cells in 21% O_2_ and a broadly similar 87% in 2% O_2_ (Figure 11a). All tendon-derived cell populations displayed increased CD90 relative fluorescence in 21% O_2_ compared with cells cultured in 2% O_2_ (Figure 11b). Therefore, we determined that type II TSCs in 2% O_2_ and TNCs in 21% O_2_ have the highest number of CD90-positive cells, while, as for CD44, TNCs in both oxygen conditions have the highest fluorescence intensity.

Unlike MSCs, all tendon-derived cell populations expressed less than 0.5% positive cells for CD73 and CD105 (Figure 12).

From flow cytometry analysis, it was also possible to obtain information on the size (FSC-A) (Figure 13a) and granularity (SSC-A) (Figure 13b) of the tendon-derived cell populations. Type I TSCs displayed a 10% increase in forward scatter area in 2% O_2_ in contrast to 21% O_2_ (*p* = 0.0024). However, the TSC I aspect ratio showed no significant difference between type I TSCs at early passages cultured in different oxygen concentrations (3.74 ± 0.78 for 21% O_2_ and 3.47 ± 0.55 for 2% O_2_, respectively) (Figure 3). Type II TSCs also displayed a 3% increase in 2% O_2_ compared with 21% O_2_ (*p* = 0.0007). This was consistent with the TSC II aspect ratio, where a significant (*p* < 0.0001) difference between the same passage cells cultured in different oxygen conditions was measured (2.89 ± 0.55 for 21% O_2_ P1 and 5.87 ± 0.94 for 2% O_2_ P1) (Figure 4). TNCs displayed a 24% increase in the forward scatter area in cells cultured in atmospheric oxygen compared with the physiological one (*p* < 0.0001). The TNC aspect ratio measurements also showed a difference (*p* < 0.0001) between cells cultured in different oxygen concentrations (2.98 ± 0.87 for 21% O_2_ P1 and 3.95 ± 0.75 for 2% O_2_ P1) (Figure 5).

Side scatter area measurement showed that type I TSCs displayed a 20% significant increase in cells cultured in 2% O_2_, and type II TSCs displayed a non-significant increase; in contrast, TNCs displayed a 24% decrease in cells cultured in 2% O_2_.

We concluded that the TNCs’ forward scatter and side scatter areas were the most affected by oxygen culture condition, displaying the highest value in 21% O_2_ and the lowest in 2% O_2_ among all tendon-derived cell populations.

### 2.6. Tendon-Derived Cell Populations Express Both Tenogenic and Stem Cell Markers

Through differentiation and immunophenotype analysis, we determined that all tendon-derived cell populations displaying multipotent potential were positive for CD44 and CD90 but not CD73 and CD105 and did not express hematopoietic markers. We next sought to establish whether each cell population retained the expression of tendon-linked gene expression during extended culturing and whether this was accompanied by canonical stem cell gene expression.

Type I TSCs displayed increased TNMD expression between early- and late-passage cells, both in 2% O_2_ (*p* = 0.0379) and in 21% O_2_ (*p* = 0.0473). A significant difference was noted between early-passage cells cultured in different oxygen concentrations, with TSC I early-passage TNMD expression in 2% O_2_ being two-fold higher than in early-passage type I TSCs in 21% O_2_ (*p* < 0.0001) and between late-passage cells (*p* = 0.0473) (Figure 14a). SCX-A expression did not change between different passages in type I TSCs in 2% O_2_, while a 1.5-fold increase was measured in 21% O_2_ (*p* = 0.0035). Moreover, early-passage TSC I SCX-A expression in 2% O_2_ was 1.4-fold higher than in 21% O_2_ (*p* = 0.0233) (Figure 14b). A 1.4-fold decrease in THBS4 expression was determined between early- and late-passage type I TSCs in 2% O_2_ (*p* = 0.0013) in contrast with a 1.8-fold increase in type I TSCs in 21% O_2_ (*p* < 0.0001). Across early- and late-passage cells cultured in different oxygen concentrations, early-passage TSC I THBS4 expression in 2% O_2_ was 1.4-fold higher than in 21% O_2_ (*p* = 0.0021), while late-passage THBS4 expression was 1.8-fold higher in 21% O_2_ (*p* < 0.0001) (Figure 14c). Tn-C expression between early- and late-passage cells displayed a 1.7-fold decrease in 2% O_2_ (*p* = 0.0041) and a 13.5-fold increase in 21% O_2_ (*p* < 0.0001). Moreover, late-passage TSC I Tn-C expression in 21% O_2_ was 5.3-fold higher than in 2% O_2_ (*p* < 0.0001) (Figure 14d).

Type II TSCs maintained TNMD expression between early- and late-passage cells within the same oxygen condition and between different oxygen culture conditions. The only significant difference noted was between late-passage cells at different oxygen concentrations, with a 1.2-fold increase in type II TSCs cultured in 21% O_2_ (*p* = 0.0500) (Figure 14a). No significant differences were observed in SCX-A expression across all the tested combinations of passage numbers and oxygen concentrations (Figure 14b). A 1.3-fold increase in THBS4 expression occurred between early- and late-passage cells in 2% O_2_ (*p* = 0.0356), in contrast to a 1.2-fold decrease in 21% O_2_. Late-passage type II TSCs in 2% O_2_ had a significantly higher 1.4-fold expression of THBS4 compared with 21% O_2_ (*p* = 0.0012) (Figure 14c). Tn-C expression between early-and late-passage cells displayed a 2.4-fold decrease in 2% O_2_ (*p* = 0.0446) and an 8.7-fold decrease in 21% O_2_ (*p* < 0.0001). Moreover, late-passage TSC II Tn-C expression in 2% O_2_ was 4.3-fold higher than in 21% O_2_ (*p* < 0.0001) (Figure 14d).

TNCs had a 2.2-fold increase in TNMD expression between early and late passage cells in 2% O_2_ (*p* < 0.0001) and a 1.9-fold increase in 21% O_2_ (*p* = 0.0002). Significant differences were noted in the same passage cells at different oxygen concentrations, with two-fold higher TNMD expression in early-passage TNCs in 21% O_2_ (*p* < 0.0001) and 1.6-fold higher in late-passage TNCs in 21% O_2_ (*p* = 0.0036) (Figure 14a). A 1.5-fold increase in SCX-A expression between late-passage TNCs cultured in atmospheric and physiological oxygen was noted (*p* = 0.0146) (Figure 14b). THBS4 expression in TNCs was consistent across different passages and oxygen concentrations (Figure 14c). Between early- and late-passage cells, Tn-C expression displayed a 1.7-fold decrease in 2% O_2_ and a 5.8-fold decrease in 21% O_2_ (Figure 14d).

From the above, we can determine that the TSC I and TNC expression profiles of the tenogenic markers were affected by the oxygen condition and passage number, while TSC II profiles were more stable over time, with limited changes.

Type I TSCs displayed consistent NANOG expression across different passages in 2% O_2_ in contrast with an 8.5-fold increase in 21% O_2_ (*p* < 0.0001). In early-passage cells, TSC I NANOG expression in 2% O_2_ was 2.2-fold higher than in 21% O_2_ (*p* < 0.0001), while late-passage cells in 21% O_2_ were three-fold higher than in 2% O_2_ (*p* = 0.0116) (Figure 15a). As seen with NANOG, no significant differences in NESTIN expression were measured between type I TSCs in 2% O_2_ at early or late passage, in contrast with a 1.4-fold increase in 21% O_2_ (*p* = 0.0115). NESTIN expression in early-passage type I TSCs in 2% was two-fold higher than in 21% O_2_ (*p* < 0.0001) (Figure 15b). No significant differences were measured for OCT-4 expression in type I TSCs in 2% O_2_, while, in contrast, a seven-fold increase was seen for 21% O_2_ between different passages (*p* < 0.0001). Similar to NANOG, the OCT-4 expression of early-passage type I TSCs in 2% O_2_ was two-fold higher than in 21% O_2_ (*p* < 0.0001), while between late passage cells, the expression in 21% was three-fold higher than in 2% O_2_ (*p* = 0.0003) (Figure 15c).

Type II TSCs displayed a consistent expression profile for NANOG (Figure 15a), NESTIN (Figure 15b), and OCT-4 (Figure 15c) across passage number and oxygen concentrations with no differences noted.

TNCs displayed a significant increase in NANOG expression between early- and late-passage cells, both in 2% O_2_ (*p* = 0.0001) and in 21% O_2_ (*p* < 0.0001) (Figure 15a). Similarly, NESTIN expression in TNCs rose over time with a 1.4-fold increase both in 2% O_2_ (*p* = 0.0139) and in 21% O_2_ between early- and late-passage cells. Early-passage TNC NESTIN expression in 21% O_2_ was 1.5-fold higher than in 2% O_2_ (*p* = 0.0026) (Figure 15b). OCT-4 expression displayed a 3.6-fold increase between TNCs at early and late passage in 2% O_2_ (*p* < 0.0001) and a 4.7-fold increase in 21% O_2_ (*p* < 0.0001). Early-passage TNCs in 21% O_2_ showed a 2.4-fold higher OCT-4 expression than in 2% O_2_ (*p* < 0.0001), while late-passage TNCs in 21% O_2_ showed a three-fold higher expression than in 2% O_2_ (*p* = 0.0089) (Figure 15c).

From this, we can see that, for the tenogenic markers, the TSC I expression profile of stem markers was the most affected by the oxygen conditions and passage number, while the TSC II profile was constant over time, without significant changes.

### 2.7. Physoxia Promotes the Maintenance of the Tenogenic Profile

Following on from our demonstrations showing that endotenon-derived type II TSCs displayed the highest proliferative potential, the retention of multipotency at late passage, an immunophenotype positive for CD44 and CD90 markers, and consistent transcript levels both for tenogenic (TNMD, SCX-A, THBS4, Tn-C) and stem (NANOG, NESTIN and OCT-4) markers, we selected this population for further evaluation. We next evaluated TNMD protein expression in type II TSCs at passage 2 over 14 days in 2% and 21% O_2_ to determine whether the oxygen culture condition could affect the tenogenic profile of tendon-derived cells over time.

Immunofluorescence displayed, in physiological oxygen, a TNMD expression decrease on day 7 followed by an increase on day 14, while in atmospheric (21%) oxygen, a decrease on day 7, which was constant until day 14 (Figure 16a), was evidenced. From IF images, we carried out a semi-quantitative analysis to determine how the mean fluorescence intensity of TNMD expression changed over 14 days. On day 1, we first measured an MFI of type II TSCs in 21% O_2_, which was 1.5-fold higher than in 2% O_2_. Then, in both oxygen conditions, we measured a significant reduction between day 1 and day 7, a 63% decrease in 2% O_2_ (*p* < 0.0001), and a 50% decrease in 21% O_2_ (*p* = 0.0003). Finally, between day 7 and day 14, we measured a 70% increase in the TNMD expression in 2% O_2_ (*p* < 0.0001), while no significant change was observed in 21% O_2_. We conclude that, after 14 days of culture, physoxia was optimal for the maintenance of the tenogenic profile of type II TSCs, with TNMD expression in 2% being 1.5-fold higher than in 21% O_2_ (Figure 16b).

## 3. Discussion

Tendons connect muscles to bones and enable movement or joint stabilization [39]. Lesions and inflammation can occur in tendons because of mechanical stress, aging, or genetic predisposition [7,40]. Tendons possess a number of differing resident cell populations that provide inherent signaling and response roles [41]. Type I TSCs reside in the paratenon, the outer membrane of the tendon, and display characteristics of vascular stem/progenitor cells [21]. Type II TSCs reside in the endotenon, a mesh of loose connective tissue that surrounds collagen bundles within a unique niche composed primarily of an abundant ECM [12], while TNCs, located between collagen fibrils and in the interfascicular matrix, represent the endogenous differentiated cell population [10]. Here, we isolated, following optimized protocols [6,35], and compared TSCs I and II and TNCs isolated and expanded in two different oxygen conditions (2% and 21%). Among the tendon-derived cell populations, we demonstrated that type II TSCs have the highest proliferative rate across time and optimal maintenance of the tenogenic profile.

In agreement with previously published protocols for tendon cell isolation, we used type II collagenase [42,43,44,45]. Type II collagenase has a unique protein structure that may enhance its enzymatic activity [46]. Indeed, the structure of the enzyme can influence its binding affinity to collagen and its ability to cleave collagen fibers effectively [47]. For TNCs, we pre-scratched the well plates to allow the cells to grow in a preferential direction and to facilitate adherence, as previously reported [48,49,50]. We wanted to not only observe the differences between different cell types but also investigate whether different oxygen conditions exerted any influence on the same type of cell.

Previous reports have compared tendon-pooled stem cells and tenocytes derived from a number of companion species, namely pigs [27], rabbits [32], horses [33], and mice [34], but, to the best of our knowledge, no studies have characterized and compared different porcine tendon-derived cell populations in different oxygen concentrations. Regarding the porcine tendon-derived populations, previous reports have described the isolation and characterization of type II TSCs from the Achilles tendon via morphology, karyotype, growth kinetics, RT-qPCR, immunostaining analysis, and multipotency potential, but no comparison with type I TSCs and TNCs has been undertaken [6]. Consistent with this study, we noted that TSC II PDT increased over time across different passages, expressed the CD90 stem cell markers, and displayed multipotency potential. A further study explored cells extracted from the paratenon, the interfascicular matrix (endotenon), and fascicle cores describing growth kinetics, morphology, immunocytochemical analysis, and multipotency potential, demonstrating that the vascularity of porcine Achilles tendon substructures is not uniform throughout the tissue, but lacked more sensitive methods to distinguish between cell populations, such as flow cytometry [27]. We reported similar results regarding the multipotency potential, showing how all tendon-derived cell populations were able to differentiate, but TNCs displayed lower adipogenic potential compared with the other cell types. However, the growth kinetics analysis did not match with our results, which showed that TNCs had the shortest PDT, whereas in the current study, we showed that across different-passage TSCs always have the shortest PDT.

Consistent with previous reports, we noted that type II TSCs were smaller and more rounded than TNCs, which were larger and more elongated [21,26,34]. In contrast to previous reports from mice [34] and horses [33], we noted that TNCs had a greater doubling time in comparison to tendon stem cell populations. However, our findings are in concordance with other studies [27,32] where tendon stem cells proliferated significantly faster than tenocytes. Furthermore, our results were also consistent with previous reports [51] that showed that physiological oxygen promotes the enhanced proliferation of human type II TSCs, which we have now extended to include demonstrations for type I TSCs and TNCs. Indeed, the TSC I proliferation rate and immunophenotype were the most sensitive to oxygen concentration. We hypothesized that this may be consequent to their location on the vascularized outer part of the tendon [22,52] and, consequently, exposure to higher oxygen levels than those found internally. Finally, we demonstrated that TNCs, compared with tendon stem cell populations, were proliferatively restricted, as was shown in a previous study [32], but instead of measuring PDT only at P2, we showed how the PDT changed over time (P1, P5, and P10). In concordance with previous studies, TNCs had the highest PDT across all the passages, but in contrast to a previous study [27], which reported the shortest PDT at P2 for paratenon-derived tendon cells, we reported the shortest PDT for type II TSCs. In our study, all tendon-derived cell populations were maintained for 120 days in culture to explore resultant changes associated with proliferative aging. Consistent with previous reports [32,36,37], we determined that the aspect ratio decreased with proliferative age. In other words, cells lost their spindle-shaped morphology and became rounder and flatter. To the best of our knowledge, no previous reports have demonstrated a role for oxygen in directing tenocyte cell morphology, as we demonstrated via aspect ratio data and reinforced with size and granularity analyses for type II TSCs and TNCs, but not for type I TSCs.

We identified no stark difference in multipotent differentiation in early-passage tendon-derived cell populations with histologically identified differentiation into bone, fat, and cartilage. However, we clearly discerned reduced levels of differentiation in TNCs, with lower adipogenic and chondrogenic differentiation. Tendon stem cell (I and II) multipotency has been previously reported for pigs [6,27], mice [21], rabbits [32], and rats [53]. However, in contrast to earlier reports detailing an absence of adipogenic differentiation in TNCs in rabbits [32] and mice [34], our findings were consistent with those, which showed that tendon-derived fibroblasts could differentiate into adipocytes [27,54]. Similar to MSCs, no definitive immunophenotypic marker exists for TSCs [36,55,56]. Our findings were consistent with previous reports [9,17,29,45] where tendon-derived cell populations were negative for CD34 and CD45 and positive for CD90 and CD44, with the highest number of CD44-positive cells in type II TSCs (approximately 90% of the cells) [21]. CD44 is a cell surface adhesion receptor that is abundantly present in numerous cancer types and regulates metastasis via the recruitment of CD44 to the cell surface [57] and has been identified as a marker for different types of cells, including TSCs [13]. Furthermore, physiological oxygen concentrations applied to type II TSCs led to the highest number of CD90-positive cells [33] compared with type II TSCs cultured in 21% O_2_, which we confirmed with 93% and 88% positive cells, respectively. Another study reported that peritenon-derived cells had a lower number of CD90 positive cells compared with the proper tendon [21] in atmospheric oxygen, which we confirmed with 73% of TSC I-positive cells compared with 88% of TSC II-positive cells, but no distinction between TNCs and type II TSCs was considered.

We determined that tendon stem cells and tenocytes expressed tenogenic and stem markers, as previously reported [34]. Our findings demonstrated that, across different passages, type II TSCs, in most of the cases, did not significantly change their transcript levels, and that TNCs had the highest expression increase in tenogenic genes (TNMD and Tn-C) and NANOG, in agreement with an earlier study [34]. However, we showed that THBS4 expression in TNCs was consistent across different passages and oxygen concentrations with no significant differences, in contrast with previous reports, which showed that TNCs had the highest THBS4 expression both in physiological [33] and atmospheric oxygen [34]. We also demonstrated that higher transcript levels of SCX-A and TNMD were present in the so-called proper tendon cells (type II TSCs and TNCs) compared with paratenon cells (type I TSCs) in atmospheric oxygen, as has already been reported [21,58]. We observed that, in type II TSCs, the transcript levels, both of tenogenic and stem markers, were maintained over extended passages, while, conversely, O_2_ and passage number affected the gene expression levels of type I TSCs and TNCs. The stable gene expression of type II TSCs provided further confirmation of the maintenance of their features across time, while the increased tenogenic gene expression of type I TSCs and TNCs could be indicative of a shift to a probable terminally differentiated state supported by the loss of proliferative potential [27,32] and a morphology change [36,37].

In general, physoxia promoted a higher expression of all tested genes in early passage TSC I, confirming that they are the most sensitive to oxygen concentration, while type II TSCs were not significantly affected in most cases by the oxygen, and TNCs showed the highest expression of most of the genes (TNMD, THSB4, Tn-C, NANOG, NESTIN, and OCT-4) in atmospheric oxygen. Moreover, aging promoted an increase in TNMD expression in both oxygen concentrations for all the tendon-derived populations: SCX-A in 21% O_2_ for type I TSCs and TNCs and in 2% O_2_ for type II TSCs; THBS4 for type I TSCs in 21% O_2_ and for type II TSCs and TNCs in 2% O_2_; Tn-C for type I TSCs in 21% O_2_; NANOG for type I TSCs, and TNCs in both oxygen concentrations and in type II TSCs in 2% O_2_; NESTIN in type I TSCs in 21% O_2_ and TNCs in both oxygen and OCT-4 in type I TSCs; and TNCs in both oxygen conditions and type II TSCs in 2% O_2_.

Lastly, to demonstrate the influence of the oxygen culture condition on the protein expression profile of cells, we measured the TNMD expression of type II TSCs across different oxygen conditions, and we observed the retention of the tenogenic profile was promoted by physiological oxygen, as previously reported [51].

Based on our results, we can say that type II TSCs, with their enhanced proliferative potential, retention of multipotency across different passages, and consistency in transcript and protein levels over time, are a promising tendon-derived cell population for applications in therapeutic and tissue engineering approaches designed to ameliorate tendon disorders.

## 4. Materials and Methods

### 4.1. Isolation and Culture of Porcine TSCs

The isolation of tendon stem cells (TSCs), both from the paratenon (I) and the endotenon (II) was performed using a modification of a previously published protocol [2]. In brief, tendon tissue was dissected from the Achilles tendons of 6-month-old pigs, sourced from a local abattoir, and washed three times in sterile phosphate-buffered saline (PBS) (LONZA^®^, Basel, Switzerland, BE17-516F) with 10% Penicillin–Streptomycin–Amphotericin B Mixture (PSA) (LONZA^®^, 17-745E), 1% Ciprofloxacin (Sigma Aldrich, Darmstadt, GE, 17850), and 1% Gentamicin (50 mg/mL Gentamicin, LONZA^®^ 17-518L), followed by a wash in Hanks’ Balanced Salt Solution (HBSS) (LONZA^®^, BE10-543F) with 2% PSA, 0.2% Ciprofloxacin, and 0.2% Gentamicin. After washing, the tissue was cut into small pieces and digested with type II Collagenase (Gibco^TM^, Grand Island, NY, USA, 17101015) overnight at 37 °C. The enzymatic activity was neutralized with fetal bovine serum (FBS) (Biosera, Aurora, CO, USA, FB-1001/500), and tissue pieces were passed through a 70 μm cell strainer (Falcon^®^, Corning, NY, USA, 352350) to yield single-cell suspensions that were then centrifuged at 350 RCF for 8 min at room temperature. The released cells were resuspended in Dulbecco′s Modified Eagle′s Medium-Low Glucose (L-DMEM) with L-glutamine and sodium pyruvate (Corning, Corning, NY, USA, 10-014-CV), 15% FBS, 1% PSA, 0.4 ng/mL epidermal growth factor (EGF) (Sigma Aldrich, SRP3027), 2.5 ng/mL basic fibroblast growth factor (b-FGF) (Sigma Aldrich, SRP4037-50UG), and 2.5 ng/mL stem cell factor (SCF) (Peprotech, Hamburg, GE, 300-07). To prevent contamination, 0.1 μg/mL mycoplasma removal agent (MRA) (Bio-rad, Feldkirchen, GE, BUF035) was added for the first two weeks.

Cells were seeded at 5200 cells/cm^2^ and cultured at 37 °C in two different oxygen conditions, namely, atmospheric (21%) and physiological (2%). After 48 h of initial plating, wells were washed twice to remove the remaining non-adherent bodies. After one week, when cells reached confluence, they were labeled as passage 0 (P0), split using 0.05% (*w*/*v*) trypsin/0.02% (*w*/*v*) ethylenediaminetetraacetic acid (EDTA) (LONZA^®^, BE02-007E) (5 g/L trypsin and 2 g/L EDTA diluted 1:10 in calcium- and magnesium-free PBS), and seeded at 5000 cells/cm^2^.

### 4.2. Isolation and Culture of Porcine TNCs

The isolation of tenocytes (TNCs) was performed using small pieces of minced tendon fiber tissue and following an explant method by plating directly into 6-well plates, allowing them to adhere for 2–3 h, and then carefully covering them with standard culture medium: DMEM High Glucose (Corning, 15-013-CV), 10% FBS, 1% PSA, 1% L-glutamine 2 mM (L-glut) (LONZA^®^, BE17-605E), and 1% Non-Essential Amino Acids (NEAAs) (LONZA^®^, 13-114E). To promote tissue adherence, 6-well plates were pre-scratched with a surgical blade. For preventive purposes, MRA was added at 0.1 μg/mL for the first 2 weeks. Culture media were changed weekly until the emergence of TNCs when tissue pieces were discarded and fresh media added. When TNCs reached confluence, they were labeled as passage 0 (P0), split enzymatically, and seeded thereafter at 5000 cells/cm^2^.

### 4.3. Cell Characterization

#### 4.3.1. Growth Rate

To determine the growth rate, cells were cultured for 120 days. Population doublings (PDs) were calculated using the formula PD = log10 (N/N0)/log10(2), where N was the number of harvested cells at the passage, and N0 was the initial number of cells. The resulting values were then used to calculate the cumulative population doublings (CPDs) as a function of time using the formula CPD_n_ = PD0 + PD_n_ + PD_n-2_ (where ‘n’ is the time point that is equivalent to the days in culture) to plot a growth curve (X-axis for time point; Y-axis for CPD).

#### 4.3.2. Morphological Analysis

Images of all cell types were captured with an inverted optical microscope (Olympus CKX4) and analyzed for shape, cell area, and aspect ratio using the manual tool in the ImageJ analysis software [59] (v 1.53c) to draw the perimeter of the cells, and then the aspect ratio was measured by dividing the major axis length of the cell area (length) over the minor axis length (width). Thirty individual cells were randomly selected within three image fields.

#### 4.3.3. Multipotent Differentiation

Trilineage differentiation (adipocytes, osteocytes, and chondrocytes) was performed as follows: cells were cultured for 21 days and seeded at 15,000 cells/cm^2^. The experiment was performed in both oxygen conditions (2% and 21%) at early passage (P3) for TSC types I and II and TNCs and late passage for type II TSCs (P12).

Adipogenic medium consisted of DMEM supplemented with 0.1µM dexamethasone (Sigma Aldrich, d2915), 0.5 mM 3-Isobutyl-1-methylxanthine (Sigma Aldrich, I5879), 10 mg/mL human insulin solution (Sigma Aldrich, I9278), 100μM indomethacin (Sigma Aldrich, I7378), 10% *v*/*v* FBS, 1% *v*/*v* NEAA, 1% *v*/*v* L-glut, and 1% *v*/*v* PSA. Osteogenic medium consisted of DMEM supplemented with 50µM ascorbic acid (Sigma Aldrich, A4544), 10 μM β-glycerophosphate (Sigma Aldrich, G9422), 0.1 μM dexamethasone, 10% *v*/*v* FBS, 1% *v*/*v* NEAA, 1% *v*/*v* L-glut, and 1% *v*/*v* PSA. Chondrogenic media consisted of DMEM supplemented with 1% *v*/*v* Insulin–Transferrin–Selenium (Gibco^TM^, 41400045), 0.1 μM dexamethasone, 50 μM ascorbic acid, 40 μg/mL L-proline, 1% *v*/*v* sodium pyruvate (Gibco^TM^, 11360070), 10 ng/mL Recombinant human TGF-β3 (Peprotech, 100-36E), 1% *v*/*v* FBS, 1% *v*/*v* NEAA, 1% *v*/*v* L-glut, and 1% *v*/*v* PSA.

Cells were seeded on day -1 in standard culture medium. After 24 h, at day 0, standard medium was replaced with differentiation media where appropriate for the day 7, 14, and 21 time points. Media were changed twice a week; cells were fixed with 10% formalin and stored at 4 °C with PBS

Histological staining was performed using 3 mg/mL Oil red O (Sigma Aldrich, O0625-100G) in 99% isopropanol, 2% alizarin red S sodium salt (Alfa Aesar, Haverhill, MA, USA, 42040) in distilled water (dH_2_O), and 1% Alcian blue 8GX (Sigma Aldrich, A3157-10G) in 0.1 M aqueous HCl. Pictures were taken with the EVOS™ XL Core Configured Cell Imager (Invitrogen^TM^, Horsham, UK, AMEX1100).

#### 4.3.4. Flow Cytometry

TSC types I and II and TNCs at P3 were detached and counted, and 150,000 cells were taken for labeling. Labeling was performed following the manufacturer’s instructions (Miltenyi Biotech, Bergisch Gladbach, GE). Briefly, 150,000 cells for each marker were spun at 300 RCF for 3 min and then incubated for 10 min at 4 °C with the following directly PE-conjugated anti-human antibodies: IgG1 (#130-119-859), IgG2 (#130-123-273), CD14 (#130-113-709), CD19 (#130-113-646), CD34 (#130-113-741), CD44 (#130-102-66), CD45 (#130-110-770), CD73 (#130-120-152), CD90 (#130-114-902), HLA-DR (#130-113-402), and CD105 (#130-112-321) (all purchased from Miltenyi Biotec). IgG1, unstained cells, and IgG2 were used as controls. After antibody incubation, samples were washed twice with PBS and resuspended in PBS for acquisition. Samples were measured with CytoFlex (Beckman Coulter, High Wycombe, UK) and analyzed with the CytExpert 2.2 software. A minimum of 20,000 events were recorded. Flow cytometry events were first gated by plotting forward scatter (FSC) vs. side scatter (SSC) and then excluding double cells (FSC-A vs. FSC-H) before determining CD surface marker expression.

#### 4.3.5. RNA Isolation and Gene Expression Profile via Quantitative Reverse Transcription PCR (RT-qPCR)

RNA extraction was performed with the RNeasy Mini Kit (Qiagen, Hilden, Germany, 74104), according to the manufacturer’s instructions, from samples collected at both early and late passage. For each cell type, RNA was extracted from samples collected at both early and late passage, and the concentration of the extracted RNA was quantified using a NanoDrop2000 spectrophotometer (ThermoFisher, Horsham, UK). Samples were then diluted with RNase-free water to a working solution of 10 ng/μL and stored at −20 °C.

RT-qPCR was undertaken using QuantiNova SYBR Green RT-PCR Kit (Qiagen, 208156), according to the manufacturer’s instructions. It was performed on an AriaMx Real-Time PCR (Agilent Technologies, Santa Clara, CA, USA) using the following cycling conditions: reverse transcription (RT) at 50 °C for 10 min and hot start at 95 °C for 2 min, followed by 40 repeated cycles of amplification at 95 °C for 5 s for denaturation and combined annealing/extension at 60 °C for 10 s. The final melt cycle was for 30 s at 95 °C, followed by 30 s at 65 °C and 30 s at 95 °C. RT-qPCR products were also visualized with ethidium bromide on 2% agarose gel electrophoresis. Primers were designed using Primer-BLAST: Scleraxis (SCX), Tenomodulin (TNMD), Thrombospondin-4 (THBS4), Tenascin C (Tn-C), Nanog, Oct-4, Nestin, and Actin β (ACTβ) were synthesized by Sigma-Aldrich (Table 1). Triplicate experiments were performed for each condition studied, and data were normalized to ACTβ.

#### 4.3.6. Immunofluorescence (IF) Analysis

IF was performed to evaluate tenomodulin (TNMD) expression in type II TSCs in different oxygen conditions (2% and 21%) over 14 days. Type II TSCs were seeded at 1500 cells/cm^2^, and standard culture media were changed twice weekly. Cells at each time point (days 1, 7, and 14) were fixed with 10% formalin, overlaid with PBS, and stored at 4 °C until use. For labeling, PBS was first discarded, and permeabilization buffer (0.15% Triton X-100 in PBS) (Sigma-Aldrich, 9036-19-5) was added for 10 min at room temperature and washed with PBS before adding a blocking buffer (1% BSA + 0.1% Tween-20 in PBS) for 1 h. After 1 h, samples were washed with PBS, and the primary antibody (anti-tenomodulin antibody) (Abcam, Cambridge, UK, ab203676) was added (1:100 dilution). Samples were incubated overnight at 4 °C. Samples were then washed three times with PBS and a secondary antibody (Goat Anti-Rabbit IgG H&L Alexa Fluor^®^ 48, pre-adsorbed) (Abcam ab150081) was added (dilution 1:500) and incubated in the dark for 1 h at room temperature. After incubation, samples were washed with PBS, and DAPI was added for 10 min at room temperature. After a final wash with PBS, samples were observed with a Nikon Eclipse Ti-S microscope, and images were captured (Nikon DSi 1 camera) using the micro-Manager microscopy software. Semi-quantitative analysis was performed with ImageJ (v 1.53 c) measurement tools for mean fluorescence intensity, selecting 35 random cells in each image and subtracting the mean gray value of the local background area.

#### 4.3.7. Statistical Analysis

Results obtained from multiple experiments (*n* = 3 biological replicates) are represented as mean ± standard deviation (SD). Student’s T-test and one-way and two-way ANOVA were applied for statistical analysis using the GraphPad Prism software (v. 9.4.1 for Windows, LLC., San Diego, CA, USA). Statistical significance is indicated by * for *p* < 0.05, ** for *p* < 0.01, *** for *p* < 0.001, and **** for *p* < 0.0001.

## 5. Conclusions

In this study, we examined the isolation, identification, and biological characteristics of three distinct porcine tendon-derived cell populations. They displayed some similarities, such as multipotency potential and immunophenotype, but also distinct properties, such as their proliferative rate and transcript levels. Among these cell populations, we selected type II TSCs for further examination, considering their enhanced proliferative potential, multipotency capacity, and the retention of their tenogenic profile over time, both in transcript and protein expression, especially in physiological oxygen conditions. Consequently, our findings suggest that porcine type II TSCs may be a promising source of stem cells for regenerative medicine therapies in animal models. Further studies will be required to understand how type II TSCs can contribute to the healing and repair of tendons following injury.

## Figures and Tables

**Figure 1 ijms-24-15107-f001:**
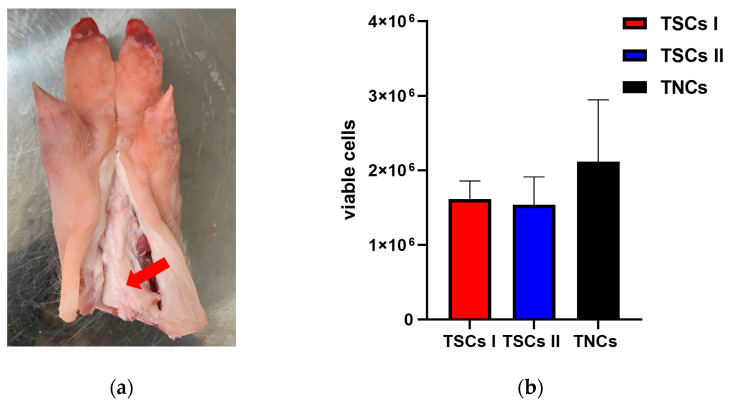
Tendon stem cells were successfully isolated from porcine Achilles tendons through enzymatic digestion. (**a**) Porcine Achilles tendon, indicated by red arrow, from which tendon cells were isolated after any visible fat or muscle were removed; (**b**) viable type I TSCs (red), derived from the digestion of the paratenon; type II TSCs (blue), derived from the digestion of the endotenon; and TNCs (black) derived from tissue explant (2% and 21% O_2_ averaged together); *n* = 3. Data expressed as mean ± SD.

**Figure 2 ijms-24-15107-f002:**
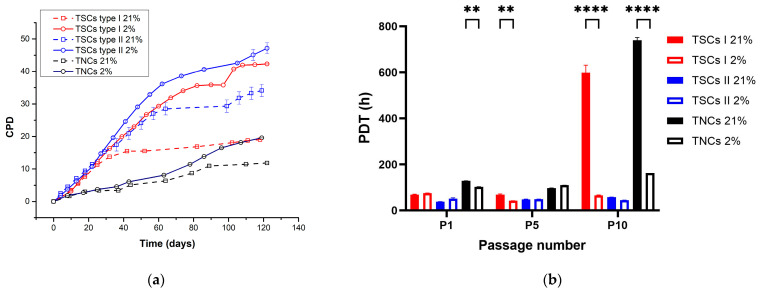
Growth kinetics of tendon cells. (**a**) Physoxia promotes higher proliferative potential in tendon cell populations. TSCs I (red) and II (blue) and TNCs (black) were isolated and cultured in 2% and 21% O_2_ to calculate cumulative population doubling (CPD); (**b**) tendon cell populations have different population doubling times. Type II TSCs (blue) have the shortest PDT during the whole culture time compared with type I TSCs (red) and TNCs (black). Data expressed as mean ± SD. ‘**’ = *p* < 0.01 and ‘****’= *p* < 0.0001.

**Figure 3 ijms-24-15107-f003:**
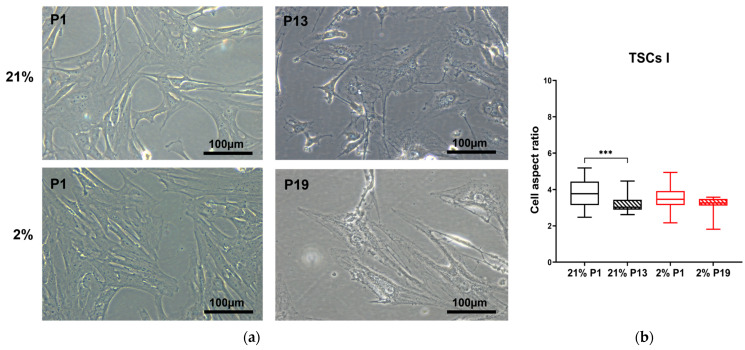
TSC I aspect ratio is oxygen-dependent. (**a**) TSC I morphology changes in 21% and 2% O_2_, observed through optical microscopy between early and late passages. Scale bar = 100 µm. (**b**) TSC I aspect ratios were measured and compared between early and late passages within the same oxygen culture condition and between the same passages at different oxygen concentrations. ‘***’ = *p* < 0.001.

**Figure 4 ijms-24-15107-f004:**
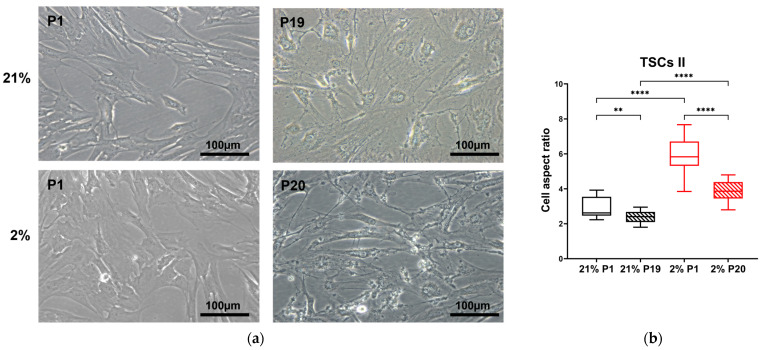
TSC II aspect ratio is oxygen- and passage-dependent. (**a**) TSC II morphology changes in 21% and 2% O_2_ observed through optical microscopy between early and late passages. Scale bar = 100 µm. (**b**) TSC II aspect ratios were measured and compared between early and late passages within the same oxygen culture condition and between the same passages at different oxygen concentrations. ‘**’ = *p* < 0.01 and ‘****’= *p* < 0.0001.

**Figure 5 ijms-24-15107-f005:**
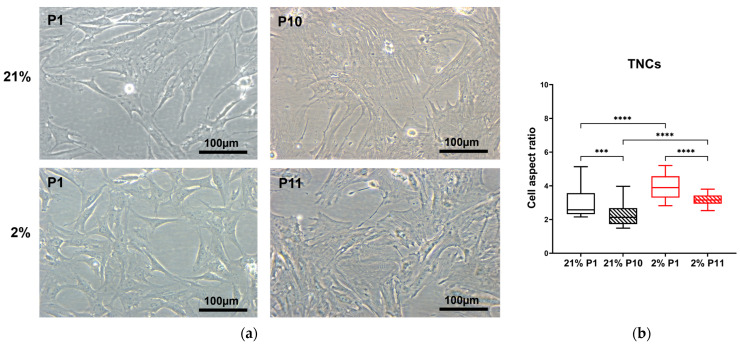
TNC aspect ratio is oxygen and passage-dependent. (**a**) TNC morphology changes in 21% and 2% O_2_ observed through optical microscopy between early and late passages. Scale bar = 100 µm. (**b**) TNC aspect ratios were measured and compared between early and late passages within the same oxygen culture condition and between the same passages at different oxygen concentrations. ‘***’ = *p* < 0.001 and ‘****’= *p* < 0.0001.

**Figure 6 ijms-24-15107-f006:**
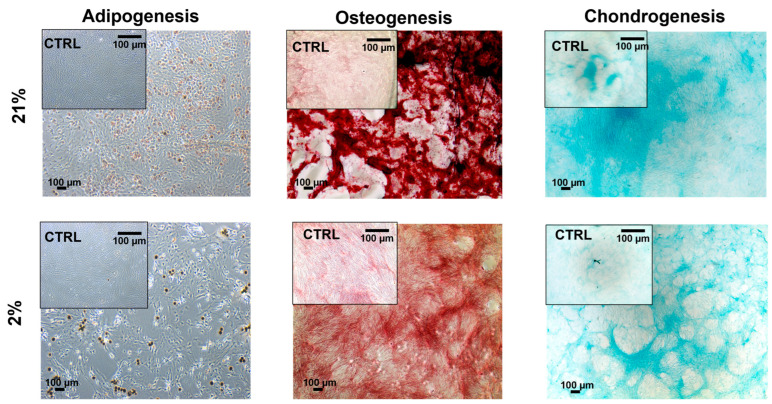
Multipotency potential of type I TSCs. Trilineage differentiation (adipogenesis, osteogenesis, and chondrogenesis) of type I TSCs at an early passage (P3) in 21% and 2% O_2_. Cells were cultured for 21 days using complete culture media as a control. Scale bar = 100 µm.

**Figure 7 ijms-24-15107-f007:**
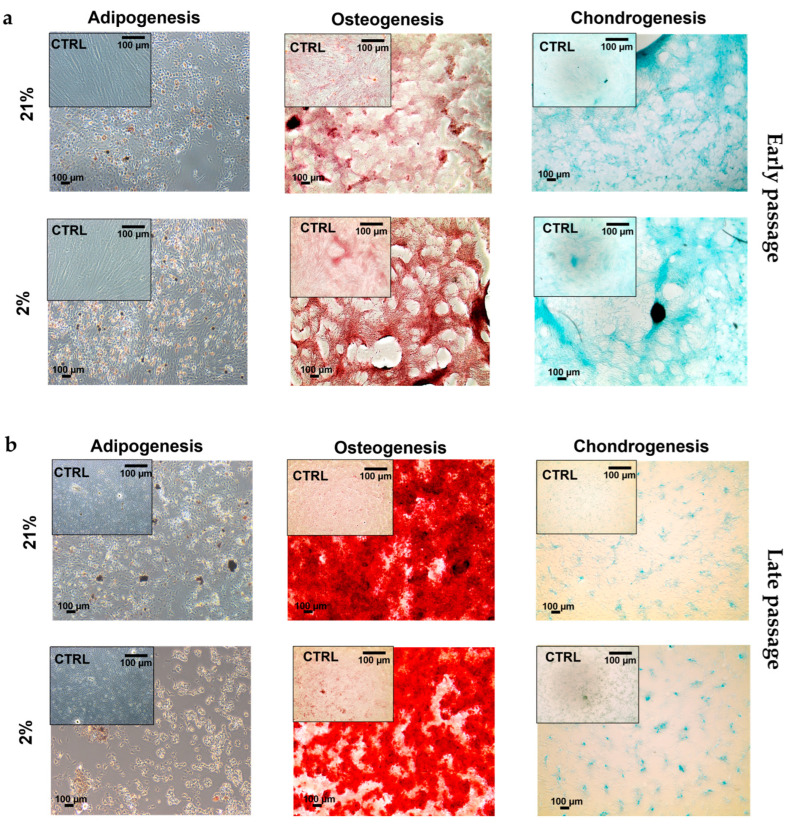
Multipotency potential of type II TSCs. Trilineage differentiation (adipogenesis, osteogenesis, and chondrogenesis) of type II TSCs at (**a**) an early passage (P3) and (**b**) a late passage (P13) in 21% and 2% O_2_. Cells were cultured for 21 days using complete culture media as a control. Scale bar = 100 µm.

**Figure 8 ijms-24-15107-f008:**
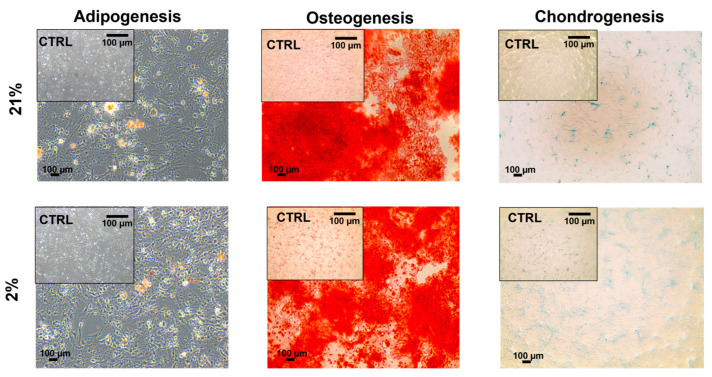
Multipotency potential of TNCs. Trilineage differentiation (adipogenesis, osteogenesis, and chondrogenesis) of TNCs at an early passage (P3) in 21% and 2% O_2_. Cells were cultured for 21 days using complete culture media as a control. Scale bar = 100 µm.

**Figure 9 ijms-24-15107-f009:**
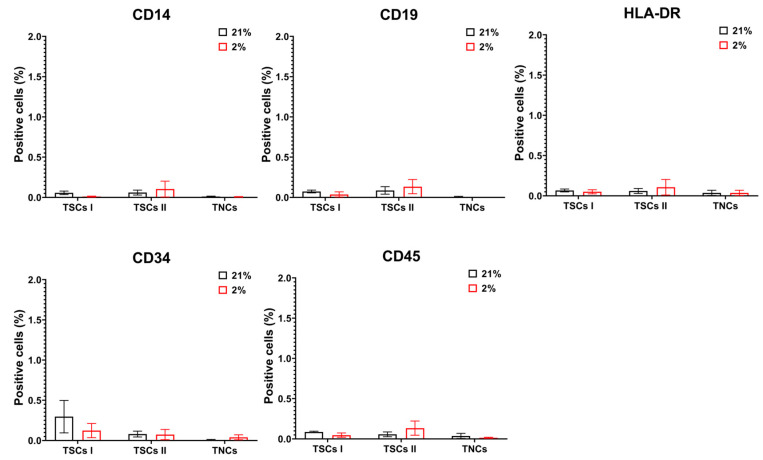
TSCs I and II and TNCs do not express hematopoietic markers. The panel shows CD14, CD19, CD34, CD45, and HLA-DR surface marker expression for all cell types through flow cytometry analysis. Cell count bar charts display marker expressions (% of positive events) on single cells.

**Figure 10 ijms-24-15107-f010:**
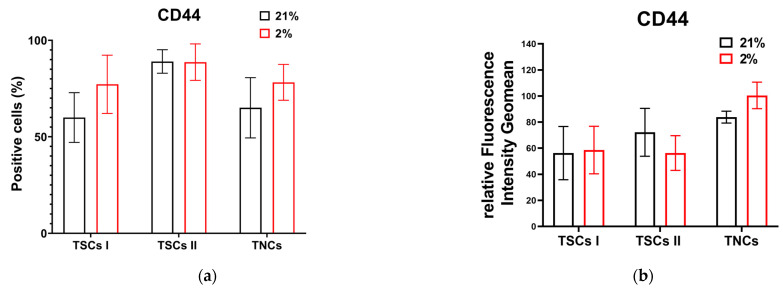
Tendon-derived cell populations expressing CD44. (**a**) Cell count bar charts display CD44 marker expressions (% of positive events) in single cells for TSCs I and II and TNCs; (**b**) relative fluorescence intensity geomean (rFIG) of all cell types that expressed CD44.

**Figure 11 ijms-24-15107-f011:**
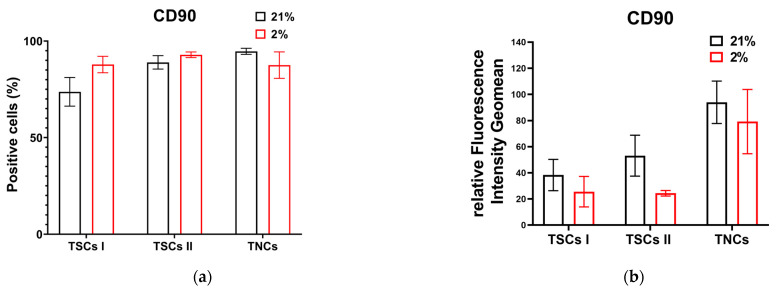
Tendon-derived cell populations express CD90. (**a**) Cell count bar charts display CD90 marker expressions (% of positive events) in single cells for TSCs I and II and TNCs; (**b**) relative fluorescence intensity geomean (rFIG) of all cell types that expressed CD90.

**Figure 12 ijms-24-15107-f012:**
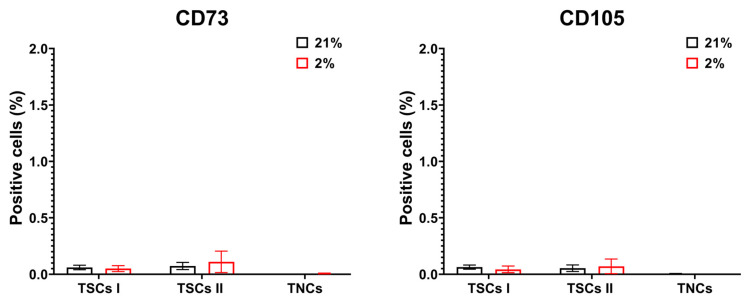
TSCs I and II and TNCs do not express CD73 and CD105. The panel shows CD73 and CD105 surface marker expression for all cell types through flow cytometry analysis. Cell count bar charts display marker expressions (% of positive events) in single cells.

**Figure 13 ijms-24-15107-f013:**
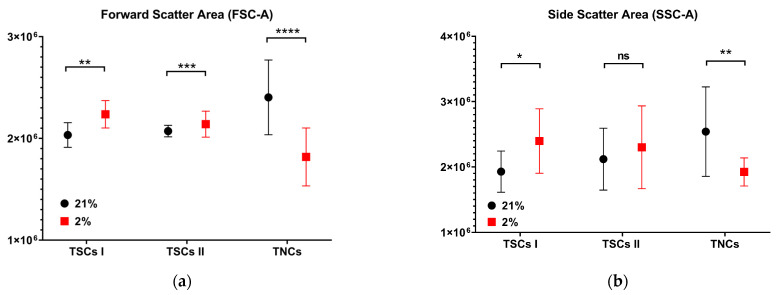
TSCs and tenocytes have distinct responses to air exposure in size and granularity. The panel shows (**a**) the forward scatter area (FSC-A) of TSCs I and II and TNCs; (**b**)the side scatter area (SSC-A) of TSCs I and II and TNCs. ‘*’ for *p* < 0.05, ‘**’ for *p* < 0.01, ‘***’ for *p* < 0.001, and ‘****’ for *p* < 0.0001, ‘ns’ indicates ‘non significant’.

**Figure 14 ijms-24-15107-f014:**
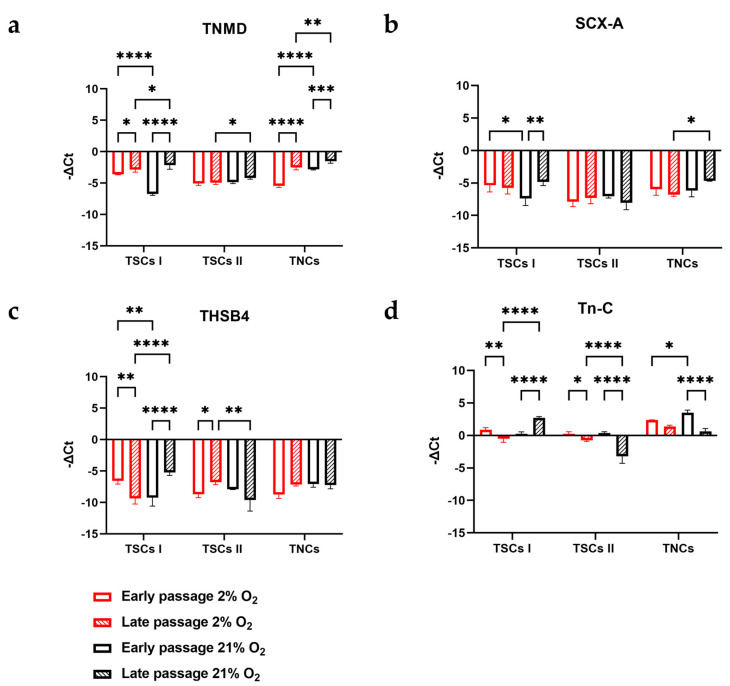
Tendon-derived cell populations express tenogenic genes. TSCs I and II and TNCs were tested at an early passage and a late passage at 2% and 21% O_2_ for the expression of (**a**) TNMD, (**b**) SCX-A, (**c**) THBS4, and (**d**) Tn-C. Data are expressed as −ΔCt, normalizing each gene of interest to ACT-β (housekeeping gene). ‘*’ for *p* < 0.05, ‘**’ for *p* < 0.01, ‘***’ for *p* < 0.001, and ‘****’ for *p* < 0.0001.

**Figure 15 ijms-24-15107-f015:**
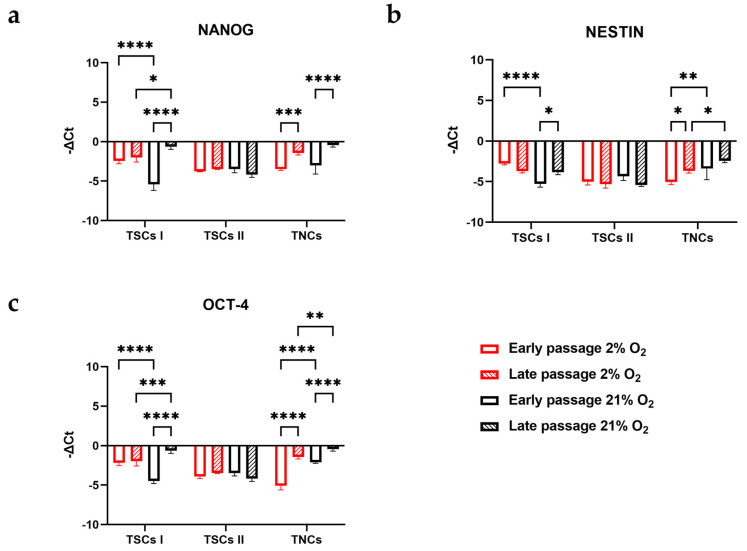
Tendon-derived cell populations expressing stem markers. TSCs I and II and TNCs were tested at an early passage and a late passage at 2% and 21% O_2_ for the expression of (**a**) NANOG, (**b**) NESTIN, and (**c**) OCT-4. Data are expressed as −ΔCt, normalizing each gene of interest to ACT-β (housekeeping gene). ‘*’ for *p* < 0.05, ‘**’ for *p* < 0.01, ‘***’ for *p* < 0.001, and ‘****’ for *p* < 0.0001.

**Figure 16 ijms-24-15107-f016:**
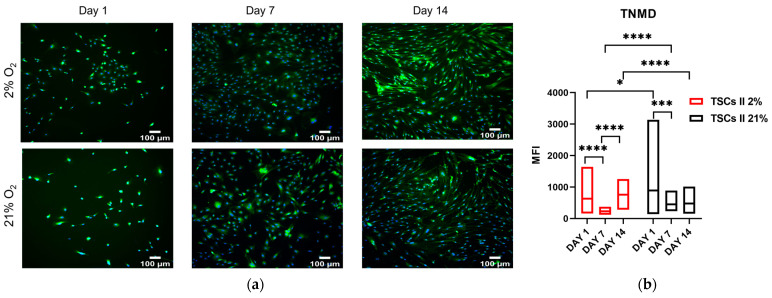
Physoxia promotes better maintenance of the tenogenic profile in type II TSCs. (**a**) Immunofluorescence images of TNMD expression at days 1, 7, and 14 of type II TSCs cultured in 2% and 21% O_2_. Scale bar = 100 µm. (**b**) MFI analysis of TNMD expression of type II TSCs cultured for 14 days in 2% and 21% O_2_. ‘*’ for *p* < 0.05, ‘***’ for *p* < 0.001, and ‘****’ for *p* < 0.0001.

**Table 1 ijms-24-15107-t001:** Primer sequences and conditions for RT-PCR.

Gene	Sequence
Scleraxis	F 5′-FCAGCAGCACCTGTAACCCAAG-3′
	R 5′-AGACTCGTGGGGACGAAGA-3′
Tenomodulin	F 5′-AGAAGACCCGTCATGCCAGA-3′
	R 5′-AAGGAGCAGTGAGTTTTGCGA-3′
Thrombospondin-4	F 5′-ACCAGGGTACATCGGGATCA-3′
	R 5′-GCAAGGGTTCAGCTCTGGAT-3′
Tenascin C	F 5′-AGATCTCGATTCTCCAAGAG-3′
	R 5′-CCGTCAACAGATTCATACAC-3′
Nanog	F 5′-CAGTGATTTGGAGGCCGTCT-3′
	R 5′-TCCATGATTTGCTGCTGGGT-3′
Oct-4	F 5′-ACCCCTCCCAGAGCTTATGAT-3′
	R 5′-CTGCTTGATCGTTTGCCCTT-3′
Nestin	F 5′-GTCTTAGTCTCAGCTCCGTGG-3′
	R 5′-GTCAGGCTGAATGGTTGGGG-3′
Actin β	F 5′-GCTAAGGGGGCGCTCTGTC-3′
	R 5′-GTGTTGGCGTAGAGGTCCTTC-3′

## Data Availability

Data available on request.

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
