# Peer review of "Endotenon-Derived Type II Tendon Stem Cells Have Enhanced Proliferative and Tenogenic Potential"

_ijms, 2023, doi:10.3390/ijms242015107_

Round 1

Reviewer 1 Report

The authors examined the isolation, identification, and biological characteristics of three distinct porcine tendon-derived cell populations.

All parts of the manuscript are well written and very good organized: introduction, results, methods, discussion and conclusion.

I have no further comments.

Reviewer 2 Report

Clerici et al. report in this paper the morphology and function of different types of tenocytes (TNCs) and tendon stem cells (TSC1 and TSC2) from porcine Achilles tendon and the impact of oxygen concentrations. Tendinopathy is a serious health problem and the data of the paper contribute to a better understanding of the cellular composition of tendons and the potential use of these cells for novel therapeutic approaches.

The following question arose during the review process:

1. The authors need to show the number of TNCs which they isolated from the tendons. The data is missing in Fig. 1b.

2. The purity of TSC1, TSC2 and TNC should be presented in the paper.

3. It is very difficult to recognize the morphology of the cells in the photos.

4. Fig. 2, the SD and statistical analysis is missing.

5. Collagen production by tenocytes is an important criteria for tendon regeneration. This aspect is missing in the paper. Do TSC1, TSC2 and TNCs produce similar amounts of Col1 and Col3? How is it influenced by oxygen concentrations?

6. The impact of oxygen concentrations on the TNCs and stem cell markers is surprisingly mild. The paper would benefit from the inclusion of classical hypo- or hyperoxia-induced genes such as HIF-1 or  HO-1, respectively, as positive controls.

Reviewer 3 Report

I appreciated the work, the method followed and the results achieved which pave the way for necessary further in-vivo investigations.

Good comparison with previous studies.

Round 2

Reviewer 2 Report

The authors have largely followed the suggestions of the first review. So the quality of the paper has significantly increased and the manuscript is now suitable for publication in IJMS.